# Novel Computational Design of Polymer Micromachined Insect-Mimetic Wings for Flapping-Wing Nano Air Vehicles

**DOI:** 10.3390/biomimetics9030133

**Published:** 2024-02-22

**Authors:** Vinay Shankar, Nagi Shirakawa, Daisuke Ishihara

**Affiliations:** Department of Intelligent and Control Systems, Kyushu Institute of Technology, 680-4 Kawazu, Iizuka 8208502, Fukuoka, Japan; shankar.vinay374@mail.kyutech.jp (V.S.); shirakawa.nagi877@mail.kyutech.jp (N.S.)

**Keywords:** flapping-wing nano air vehicles (FWNAVs), 2.5-dimensional (2.5D), insect-mimetic micro wing (IMMW) model, camber, design window (DW), polymer micromachining

## Abstract

The flapping wings of insects undergo large deformations caused by aerodynamic forces, resulting in cambering. Insect-mimetic micro wings for flapping-wing nano air vehicles mimic these characteristic deformations. In this study, a 2.5-dimensional insect-mimetic micro wing model for flapping-wing nano air vehicles is proposed to realize this type of wing. The proposed model includes a wing membrane, a leading edge, a center vein, and a root vein, all of which are modeled as shell elements. The proposed wing is a 2.5-dimensional structure and can thus be fabricated using polymer micromachining. We conducted a design window search to demonstrate the capabilities of the wing. The design windows, which are areas of desirable design solutions in the design parameter space, are iteratively searched using nonlinear finite-element analysis under quasi-steady aerodynamic modeling. Here, thickness is selected as a design parameter. The properties of real insects, polymer materials, and fabrication conditions are used to determine the other parameters. A fabricable design solution that generates sufficient camber is found from the design windows.

## 1. Introduction

Insect flight is well known for its capabilities of hovering and excellent agile maneuvering [1], which have evolved over time [2,3]. These capabilities have been replicated to realize flapping-wing air vehicles. In 2005, the Defense Advanced Research Projects Agency (DARPA) started the Nano Air Vehicle Program, whose aim is “to develop and demonstrate an extremely small (less than 7.5 cm), ultralightweight (less than 10 g) air vehicle system with the potential to perform indoor and outdoor military missions” [4]. Several studies on nano air vehicles have been published after the start of this project [5,6,7,8,9]. Following nano air vehicles, pico air vehicles are described as having a maximum takeoff mass of 500 mg and a maximum dimension of 5 cm [10], which lies within the majority of flying insects’ ranges [11]. A prototype pico air vehicle was developed by the Harvard RoboBee project [12]. Since pico air vehicles are a subset of nano air vehicles, in the present study, we refer to all air vehicles that mimic insect flapping flight as flapping-wing nano air vehicles (FWNAVs). 

When an insect flaps its wings, its wings camber [1,2,11,13,14]. Considering that their wings are made of membrane and veins instead of internal muscles, it appears that such cambering is an outcome of the wings’ flexible deformation triggered by aerodynamic forces [14]. Camber increases the aerodynamic performance of insect flapping flight [15,16,17,18,19,20]. Hence, it is important for FWNAVs to produce a sufficient camber in their flapping wings. Here, our objective is to develop insect-mimetic micro wings (IMMWs) for FWNAVs whose camber is equivalent to that of actual insect wings.

Various insect-wing models, from basic conceptual models [21] to realistic models, have been developed [13]. Realistic models replicate structural details using finite elements [22,23]. Insect wings have a wide variety of structures, ranging from the macroscopic complexity of veins and corrugated structures [24,25] to the microscopic complexity of veins [26,27]. Models that include all of these structures could produce an IMMW that can directly reproduce a camber that is equivalent to that of actual insect wings. However, no such model wings have yet been reported. Furthermore, the fabrication of such model wings using conventional fabrication techniques will be quite challenging because of their geometrical complexity and small size. Here, we overcome these difficulties by introducing a novel 2.5-dimensional (2.5D) IMMW model that can be fabricated using photosensitive adhesive polyimide membranes and sheets using polymer micromachining [28]. This technique’s primary challenge stems from the precursor PI material’s high mechanical and chemical sensitivity. The fabrication procedure must be developed following each design solution. We have previously created a micro wing that is veinless and extremely similar in size to microscopic insect wings [29]. We have also been developing a fabrication technique and process for micro-transmissions that convert a piezoelectric bimorph actuator’s small translational displacement to a large stroke angle at a sufficient level of frequency [30]. These transmissions will be used to assess the performance of various micro wings.

In the present study, we propose an IMMW model for FWNAVs based on shape simplification modeling [31]. In our previous study [31], we focused on the demonstration of the fundamental validity of this modeling. In contrast, the current study focuses on finding fabricable design solutions that can reproduce camber comparable to actual insects using a DW search, and demonstrating actual fabrication using micromachining of polymer materials (polyimide sheets, Toray Industries, Inc., Tokyo, Japan). The model can produce significant cambering that is comparable to that of actual insect wings. The novelty of the proposed IMMW model is the shape simplification using a 2.5D structure. The model consists of a wing membrane, a leading edge (LE), a center vein (CV), and a root vein (RV), all of which are modeled as shell structures. Polymer micromachining will be employed to fabricate the modeled novel wing. Here, we perform a design window (DW) search for the proposed IMMW model to demonstrate the capabilities of the wing. The DWs, which are areas of desirable design solutions in the design parameter space, are iteratively searched using nonlinear finite-element analysis under quasi-steady aerodynamic modeling. Thickness is selected as a design parameter and the other parameters are determined by the properties of real insects, polymer materials, and fabrication conditions. The design solutions from the DWs give a camber that is comparable to that of actual insect wings. We then determine the fabricable final design solution from the DWs.

The novelty of this study is summarized as follows: The proposed IMMW model has a complete 2.5D structure, fabricable using only a polymer micromachining technique based on photolithography, and at the same time, it can produce a sufficient camber comparable with that of real insects. The most recent literature review [31,32,33] strongly supports that there are no such model wings of which the length scale is equivalent to that of actual small insects, and the microfabrication of such wings has also not been reported.

## 2. Our 2.5-Dimensional Insect-Mimetic Micro Wing Model Fabricable Using Polymer Micromachining

Camber deformation in an insect’s flapping wings is schematically shown in Figure 1. As shown, the ratio of the wing chord length $C_{\alpha \beta }$ to the height of the deformed wing surface $C_{o\gamma }$ determines the extent of the camber. The proposed IMMW model, shown in Figure 2a, is composed of the wing membrane and veins, all of which are modeled as shell structures. The *y* axis and flapping axis are in line. The LE draws a stroke plane with the stroke angle Φ as it flaps, translating in the *xz* plane. The IMMW model has a 2.5D structure. Hence, it can be fabricated using a polymer micromachining technique based on photolithography using photosensitive adhesive PI membranes and sheets [28]. White, red, blue, and green represent the wing membrane, LE, CV, and RV (corresponding to their respective polymer material properties). Each vein and wing membrane material characteristic is derived from PI materials. The IMMW model mimics the vein network in actual insects, allowing it to produce a significant positive camber that is similar to the wings of real insects. The dipteran insect wing is considered for the proposed wing design [34]. 

In this study, we present feasible design solutions with significant positive cambering that can be fabricated using polymer micromachining. The flapping motion of the IMMW model is shown in Figure 2b. In real insects, the camber is most significant at the center of each half-stroke [35]. The aerodynamic pressure is the most significant force acting on the insect wing, along with other aerodynamic forces [36]. As a result, this research takes into account the camber caused by aerodynamic pressure. A trapezoidal function is used as a flapping angular velocity model [37] based on observations of flapping insect wings [1]. The dynamic pressure obtained using the quasi-steady aerodynamic model is given as follows:(1)$$P=\frac{1}{2}C_{D}\rho^{f}V_{{max}^{2}}$$
(2)$$V_{max}=\omega_{max}r=\frac{8\Phi r}{3T_{\varphi }}$$
where *C*_D_, *ρ*^f^, *V*_max_, r, and *T_φ_* are the drag force coefficient for a flat plate, fluid mass density, maximum flapping speed at each point on the wing surface, distance from the flapping axis (*y* axis) along the wing length, and flapping period, respectively. The acceleration and deceleration time *t*_a_ is specified to the typical value of *T*_φ_/8 [37]. Since the dynamic pressure causes a large deformation, we use the finite-element method considering the geometrical nonlinearity. Equations (1) and (2) for the dynamic pressure include lift, drag, and moment. Initially, the wing has no feathering angle; therefore, we use the drag force coefficient. The present pressure is overestimated but it is comparable with the pressure for the actual angle. The gap will be filled by tuning other parameters such as the flapping frequency and the stroke angle. The wing inertia is dominant in the stroke reversals [37], while the dynamic pressure is most prominent in the middle of each half-stroke. The cambering, which is the primary design characteristic in this study, is maximum at the middle of half-stroke [35]. Hence, we used the dynamic pressure. The flexural rigidity *G* can be expressed as
(3)$$G=EI,\;I={wt}^{3}/12,$$
where *E* is the elastic modulus, and *I* is the second moment of area. w and t are the width and thickness of the IMMW model, respectively.

## 3. Nonlinear Finite-Element Solution Procedure for Shell Structures with Large Deformation for the Insect-Mimetic Micro Wing Model

The proposed IMMW model, similar to insect wings, exhibits large elastic deformation due to dynamic pressure. A nonlinear finite-element solution procedure is used to precisely predict its behavior. In this study, we use a procedure based on the total Lagrangian formulation for the geometrical nonlinearity. Here, we briefly present the nonlinear framework for this procedure in a general form. The fundamental challenge in a general nonlinear analysis is to find a body’s equilibrium state that corresponds to the applied loads. Assuming that external loads are applied as a function of time, a system of the finite elements’ equilibrium conditions can be written as
(4)R−t+∆tF=0,t+∆t
where the vector Rt is the externally applied nodal point forces in the configuration at time t, and the vector Ft is the nodal point forces that correspond to the element stresses. In an incremental step-by-step solution, the basic methodology involves presuming that the discrete time t is known and requires a discrete time t+∆t solution, where ∆t is an appropriate time increment. Since the solution is known at time t, we can write
(5)F=0,t+∆t=Ft+∆F,
where ∆F is the increment in nodal point forces from time t to time t+∆t that corresponds to the increments in element displacements and stresses. Concerning the geometric and material conditions at time t, the vector can be approximated by a tangent stiffness matrix Kt:(6)∆F=K∆Ut
where U is a vector of incremental nodal point displacements and
(7)Kt=∂Ft∂Ut

Therefore, the tangent stiffness matrix represents the derivative of the internal element nodal point forces Ft with respect to the nodal point displacements Ut. Substituting Equations (5) and (6) into Equation (4) yields
(8)K∆Ut=R−t+∆tFt

By solving ∆U, we can estimate the displacements at time t+∆t: (9)U=0,t+∆tUt+∆U

Equation (9) approximates displacements at t+∆t for applied loads using Equation (6) and by iterating until an adequate precise solution of Equation (4) is obtained using the Newton–Raphson method, extending the incremental technique in Equations (8) and (9). The incremental solution is calculated using the known total displacements instead of the displacements at the time t, after the increment in nodal point displacements. The equations used in the Newton–Raphson iteration for i=1, 2, 3, …, are
(10)K(i−1)t+∆t∆U(i)=R−t+∆tF(i−1)t+∆t
(11)U(i)t+∆t=U(i−1)∆U(i)t+∆t
under the inceptive order
(12)U(0)t+∆t=Ut, K(0)=t+∆tKt,F(0)t+∆t=Ft.

In the first iteration, the relations in Equations (10) and (11) reduce to those in Equations (8) and (9), respectively. Thereafter, iterations use the most recent estimates for nodal point displacements to evaluate element stresses, nodal point forces F(i−1),t+∆t and the tangent stiffness matrix K(i−1)t+∆t. The minimum time step used is 10^−15^ and the total time is a unit in the present nonlinear finite-element procedure.

## 4. Numerical Implementation of the Insect-Mimetic Micro Wing Model

Figure 3 shows the finite-element implementation of the proposed IMMW model using linear triangular shell finite elements. The span-wise length *L*_w_ and the chordwise length *C*_w_ of the wing are 0.0113 and 0.00311 m, respectively [31]. These dimensions were taken from a real insect wing. The material used for the wing membrane was a photosensitive adhesive PI membrane (Toray Industries, Inc., Tokyo, Japan; Young’s modulus *E*: 3.5 GPa, Poisson’s ratio: 0.49). The material used for the veins was a photosensitive adhesive PI sheet (Toray Industries, Inc., Tokyo, Japan; Young’s modulus *E*: 2.5 GPa, Poisson’s ratio: 0.49). The thickness and width of the LE, CV, and RV were chosen as the design parameters (see Section 5.1). Initially, the thicknesses of the wing membrane, LE, CV, and RV were set to 2, 736.8, 29.62, and 327.1 μm, respectively, and the widths of the LE, CV, and RV were set to 48 μm. The lengths of the leading edge and veins were specified based on *L*_w_ and *C*_w_. The initial values of their sectional dimensions were given based on Equation (3) with the flexural rigidity *G* of real insect wings [13,22], where the flexural rigidity *G* for LE, CV, and RV are 4.06 × 10^−6^ Nm^2^, 2.6 × 10^−10^ Nm^2^, and 3.5 × 10^−7^ Nm^2^, respectively. Hence, the torsional stiffness, which ranges from 1.1 × 10^−6^ to 5.2 × 10^−6^ Nm/rad [13,38], was not controlled because it was determined by these already-determined parameters. However, a torsional spring can be added to control the torsional stiffness independently [12], which will be our future work. The nonlinear finite-element solution procedure described in Section 3 was used for analyzing the large deformation of the IMMW model under the dynamic pressure given in Section 2. The commercial software MSC Marc 2015 was used. As shown in Figure 3, the left extremity of the LE was fixed. The aerodynamic pressure given by Equation (1) statically acts on the wing surface. The parameters in these equations were given as *C*_D_ = 1.19, which corresponds to that for a flat plate with an aspect ratio of 4, *ρ*^f^ = 1.2 × 10^−3^ g/cm^3^, Φ = 180°, *T_φ_* = 1/*f_φ_*, and *f_φ_* = 161 Hz, which are based on actual insect data [1].

## 5. Computational Design Based on Design Windows

### 5.1. Design Window Search Approach

Fabricability is rarely formulated as the design problem, which severely limits the viability of the design solution for the suggested IMMW model. Presenting DWs to the designer instead of a single optimal design solution allows the designer to take fabricability into account in the development of the final design solution. In this study, an iterative DW search was conducted for the LE, CV and RV, where the average camber was chosen as the design characteristic and the thickness was chosen as the design parameter. The widths of the center and root veins were tuned in order to improve the fabricability. In the DW search, the actual insect camber was the design criteria, that is, a camber larger than approximately 4%, which is similar to that of a real insect, was considered to be satisfactory. The camber was considered from 25% to 100% for the membrane and veins along the *x*-direction due to the design restriction [35].

### 5.2. Structural Design for IMMW Model

The deformation of the IMMW model for the analysis setup using the nonlinear finite-element solution procedure is depicted in Figure 4. A positive camber is obtained with the polymer material properties. Let us consider this result as the initial point of our design solution.

An iterative DW search was conducted [39] in order to find fabricable design solutions for the IMMW model. It was crucial to select a wing membrane thickness that is close to the parameters of real insects such that it could be fabricated using polymer micromachining. Within our laboratories, we can fabricate PI membranes with a thickness of approximately 4 μm, which is close to the thickness of an actual insect wing membrane. A wing membrane thickness of 4 μm was thus considered in the DW search. The thicknesses of the LE, CV, and RV were calculated as 738, 37.6, and 215 μm, respectively. PI sheets with a thickness of 40 μm (Toray Industries, Inc., Tokyo, Japan) were used for the fabrication of veins. Hence, the thicknesses were set to be multiples of 40 μm PI sheet.

#### 5.2.1. Center Vein

The CV thickness was iteratively searched while those of the membrane, LE, and RV were kept constant. Figure 5 shows the average camber versus various thicknesses of the CV. The DW for the CV thickness ranged from 88.8 to 209 μm, which corresponds to a flexural rigidity of *G* = 8.37 × 10^−9^ to 9.87 × 10^−8^ Nm^2^. Taking into account the restriction of the thickness of each PI sheet (40 μm), we chose design solutions from the DW in Figure 5, as shown in Table 1. A thickness of 120 μm gives the largest camber among the design candidates. The feathering angle θ for this thickness is closest to 45°, which gives the maximum lift for flapping insect wings [36]. Furthermore, this thickness gives the best fabricability since it corresponds to three layers of the PI sheet, which is the smallest number of layers among the candidates. Fabricability increases as the number of laminations decreases. The design solution with a center vein thickness of 120 μm was thus used for further DW searches.

#### 5.2.2. Leading Edge

The design solution of the wing membrane and the CV was derived from the previous section. For this section, with a wing membrane thickness of 4 μm, a center vein thickness of 120 μm, and the root vein thickness, the leading edge thickness was iteratively searched. Here, we imposed the sufficient suppression of the LE deflection as *u*/*L*_w_ < 1%, where *u* is the tip deflection of the LE and *L*_w_ is the span-wise length of the IMMW model, since the LE is the wing’s major supporting structure. The LE thickness varied from 280 to 1000 μm, as shown in Figure 6. As shown, the average camber is 3.5% or more when the thickness is 400 μm or more. A thickness of 600 μm is necessary to satisfy the above suppression condition, leading to a poor width-to-thickness aspect ratio of the section of 1/12 (48 μm/600 μm). To avoid this issue, the width-to-thickness aspect ratio was set to 1 while maintaining the stiffness, leading to a width and a thickness of 320 μm. Following this change, the average camber slightly decreased from 3.94% to 3.89%. Hence, a thickness of 320 μm, which corresponds to eight layers of the PI sheet, was chosen as the best design solution.

#### 5.2.3. Root Vein

In the above sections, we found the design solution for the wing membrane, CV, and LE. For this section, with the values of the previous design solutions, we found the RV design solution by varying the thickness. Figure 7 shows the average camber of the RV. As the thickness increases, the camber initially increases and then plateaus.

The DW area that gives a sufficient camber is 160 μm or more. Table 2 shows the design candidates. A thickness of 240 μm gives the highest camber without significantly affecting wing mass. Hence, a thickness of 240 μm was chosen as the best design solution for the root vein.

### 5.3. Final Design Solution for the Insect-Mimetic Micro Wing Model Taking into Account Fabrication Conditions

The best design solution for the IMMW model in Section 5.2 has thicknesses of 4, 120, 320, and 240 μm for the wing membrane, center vein, leading edge, and root vein, respectively, with a mass of 1.98 mg. It replicates the characteristics of actual insect wings, with a maximum camber of 4.23% and a feathering angle of 12.35°. The materials used for the membrane, center vein, leading edge, and root vein were a photosensitive adhesive PI membrane and sheet. Hence, the polymer micromachining process is applicable to this design solution.

However, prototyping the design solution for the IMMW model in Section 5.2 revealed that the center and root veins were detached from the wing membrane during the second-layer lamination process because of the narrow width of the veins, as shown in Figure 8a. On the contrary, there was no detachment of the LE during the complete eight rounds of laminations because of its sufficient width, as shown in Figure 8b. The width of the veins should be comparable to that of the LE. Hence, we tuned the width of the CV and RV taking into account this fabrication condition.

Figure 9a shows the mesh for the design solution in Section 5.2 and Figure 9b shows the mesh for the tuned design solution. In the tuned design solution, the widths of the center and root veins were set to 400 μm, which is close to the width of the leading edge. The thicknesses of the veins were determined using Equation (3) such that the flexural stiffness was maintained. The obtained thicknesses of the center and root veins were 60 and 120 μm, respectively. The maximum camber values for the design solution in Section 5.2 and the tuned design solution are 4.23% and 4.16%, respectively. Hence, the tuned design solution works well. Finally, we determined the thickness of the LE, CV, and RV to be 320, 80, and 120 μm, respectively, which are multiples of the PI sheet thickness (40 μm). The final design solution has LE, CV, and RV thicknesses that correspond to eight, two, and three layers of PI sheets, respectively.

Figure 10 shows the deformation of the final design solution and Figure 11 shows the camber distribution relationships between the location of the wing and the camber, which provides the wing camber at different wing sections for the best design solution in Section 5.2, the tuned design solution, taking into account sheet adhesion, and the final design solution, taking into account the requirement that the thicknesses must be multiples of 40 μm in addition to sheet adhesion. The final design solution gives a maximum camber of 4.3%, a feathering angle of 11.13°, and a wing mass of 2.05 mg. Hence, the final design solution satisfies the objective of the IMMW model to obtain the maximum camber. The main objective of this study is to achieve sufficient camber for the fabricable design which is similar to that of a real insect wing. However, the feathering angle of our wing is lower than that of an actual insect. In the actual implementation for insect-mimetic FWNAVs in our future work, the feathering angle will be increased by increasing the flapping frequency of the wing. Furthermore, according to morphological studies [40], the base of the wing has a high degree of torsional flexibility. This type of flexibility can be implemented as the base spring for our wing design, since this type of spring has been used in previous research [38]. In this way, the feathering angle can be tuned closer to that of an actual insect. 

## 6. Concluding Remarks

A novel IMMW model for FWNAVs made up of shell structures and having a 2.5-dimensional structure was presented in this work. This model can be fabricated using polymer micromachining. An iterative DW search was conducted on a wing comprising a membrane supported by the LE, CV, and RV, which represented a network of veins, to find a design solution that meets the design objective. In the DW search, the thickness was taken as the design parameter and the other parameters were determined using the characteristics of real insects and polymer materials (photosensitive adhesive PI membrane and sheets for fabricability using polymer micromachining) and the fabrication conditions. In the evaluation of the camber as a design characteristic, the aerodynamic force was modeled using the quasi-steady assumption and the nonlinear finite-element procedure, employing the total Lagrangian formulation. The satisfactory design solution gives a camber that is comparable to that of actual insects.

The best design solution for the IMMW model gives thicknesses of 4, 120, 320, and 240 μm for the wing membrane, center vein, leading edge, and root vein, respectively, with a mass of 1.98 mg, and replicates the characteristics of actual insect wings, with a maximum camber of 4.23% and a feathering angle of 12.35°. However, during prototyping, the CV and RV detached from the wing membrane during the second-layer lamination process. There was no detachment of the leading edge during the complete round of laminations because of its sufficient width. The width of these veins should thus be comparable to that of the leading edge. Hence, we tuned the width of the center and root veins without changing the flexural stiffness, taking into account this fabrication condition. The tuned design solution gives a camber similar to that of the original design solution.

Based on the tuned design solution, the DW search gave a final design solution with wing membrane, leading edge, center vein, and root vein thicknesses of 4, 320, 80, and 120 μm, respectively, the latter three of which correspond to eight, two, and three layers of the PI sheet, respectively. This solution gives a maximum camber of 4.3%, a wing mass of 2.05 mg, and a feathering angle of 11.13°. Hence, the final design solution satisfies the objective of the proposed IMMW model. The DW search gives promising fabricable design solutions. In future work, this IMMW model will be fabricated and experimentally tested. Furthermore, we have been preparing to flap the proposed wing using a micro-transmission. We have already developed the micro-transmission at this point [30]. Hence, a performance test will be conducted using this micro-transmission in our future work.

## Figures and Tables

**Figure 1 biomimetics-09-00133-f001:**
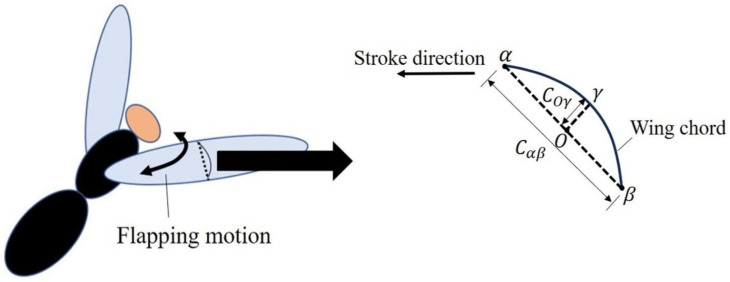
Schematic diagram of camber deformation of insect wings given by flapping motion.

**Figure 2 biomimetics-09-00133-f002:**
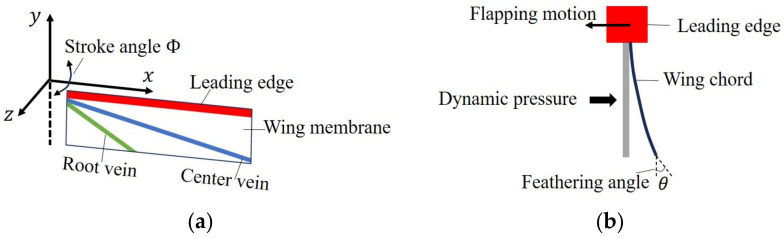
Schematic view of (**a**) an insect-mimetic micro wing (IMMW) consisting of wing membrane and veins made of shell structures and (**b**) the flapping motion.

**Figure 3 biomimetics-09-00133-f003:**
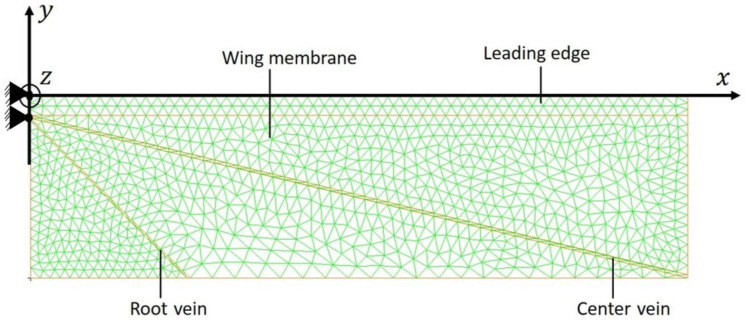
IMMW implemented using linear triangular shell finite elements with 1005 nodes and 1872 elements which are green in color represents wing membrane, while the leading edge is represented as wider red lines, center and root veins represented by narrow red lines.

**Figure 4 biomimetics-09-00133-f004:**
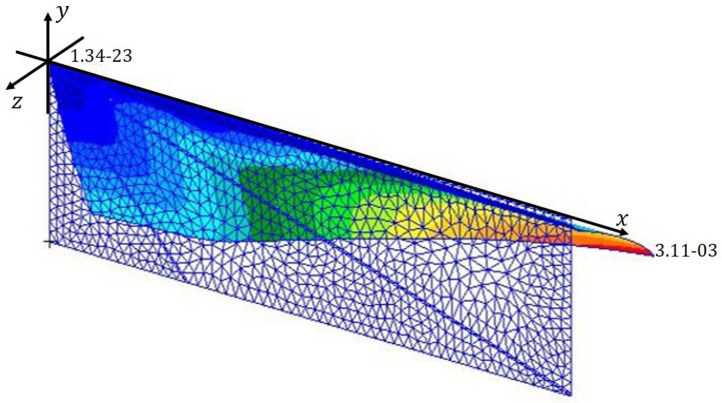
Deformation of the IMMW model for analysis setup. The color profile shows the magnitude of the displacement in the z-direction.

**Figure 5 biomimetics-09-00133-f005:**
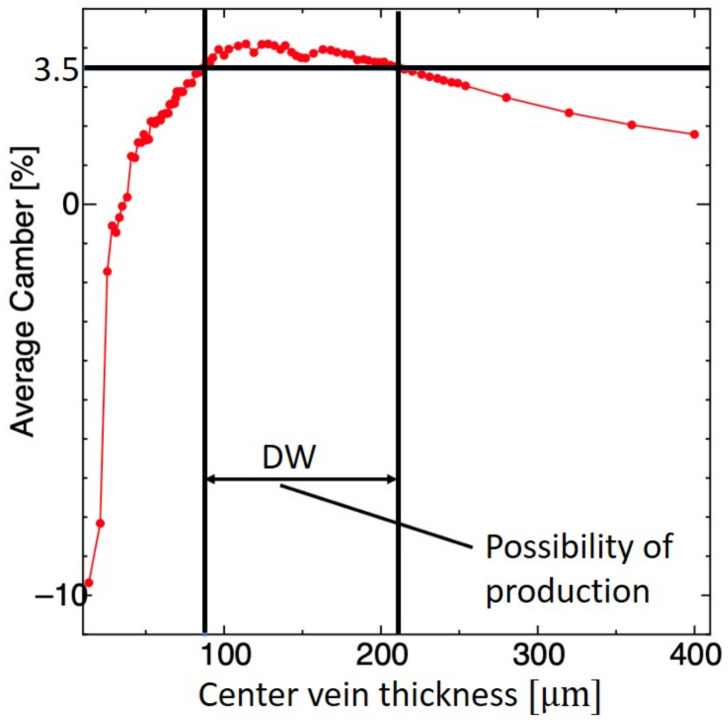
Design window (DW) search for CV thickness based on average camber. The red color indicates the average camber based on the mesh shown in Figure 3 and black horizontal and vertical lines indicates the possible design solution.

**Figure 6 biomimetics-09-00133-f006:**
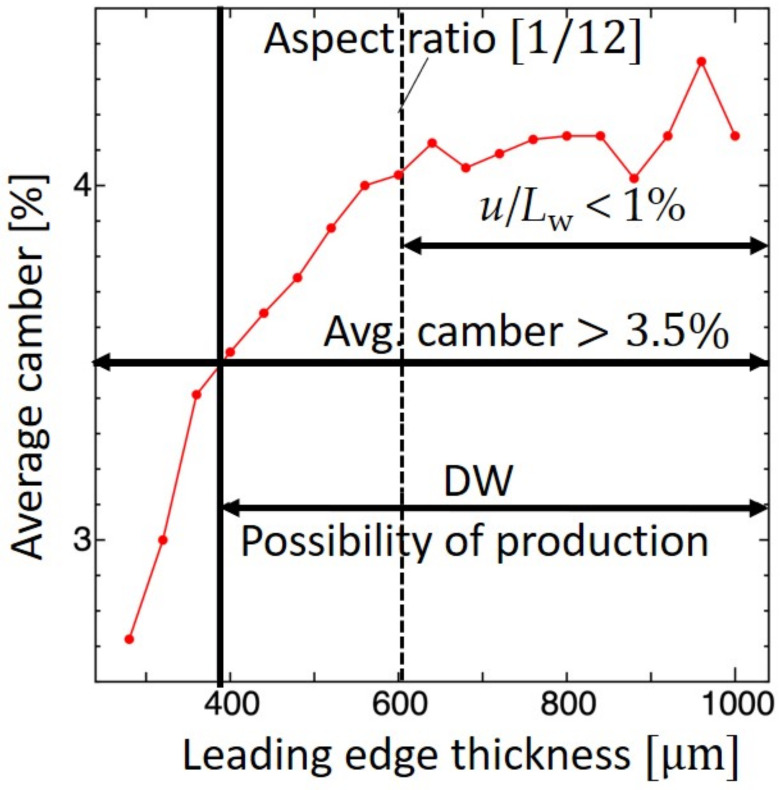
DW of leading-edge thickness versus average camber. The red color indicates the average camber based on the Figure 3 and the black dotted line indicates the aspect ratio for choosing the optimal thickness.

**Figure 7 biomimetics-09-00133-f007:**
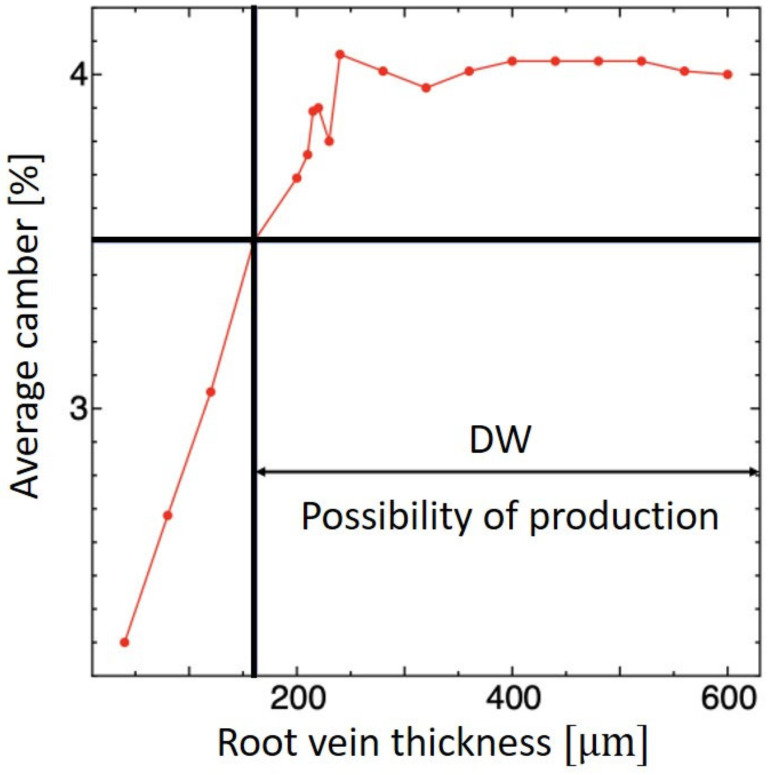
DW of root vein thickness versus average camber. The red line indicates the average camber based on the mesh shown in Figure 3 and the black line indicates the area from which the possible design solution can be chosen.

**Figure 8 biomimetics-09-00133-f008:**
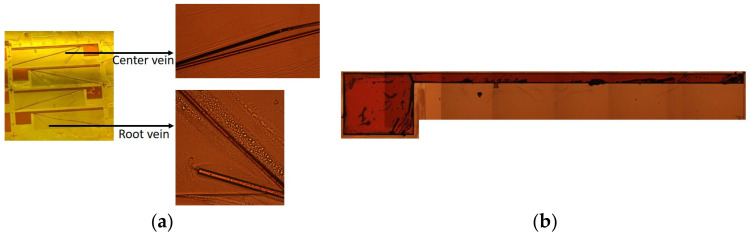
Fabrication of (**a**) the center vein and the root vein, which failed during the second layer of lamination due to a narrow width of 48 μm, and (**b**) the leading edge, with a wider width of 320 μm, which was successful.

**Figure 9 biomimetics-09-00133-f009:**
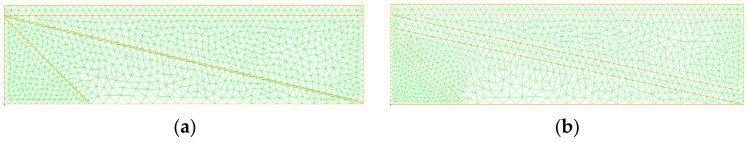
Meshes for (**a**) the original design solution with 1005 nodes and 1872 elements (see Section 5.2) and (**b**) the tuned design solution with 1053 nodes and 1956 elements, taking into account fabricability. The mesh (**a**) green color indicates wing membrane, wider red lines indicate leading edge, and narrow red lines represent center vein and root vein. The mesh (**b**) wing membrane and leading edge color indication is similar as mesh (**a**) while the wider center and root veins are indicates by red lines.

**Figure 10 biomimetics-09-00133-f010:**
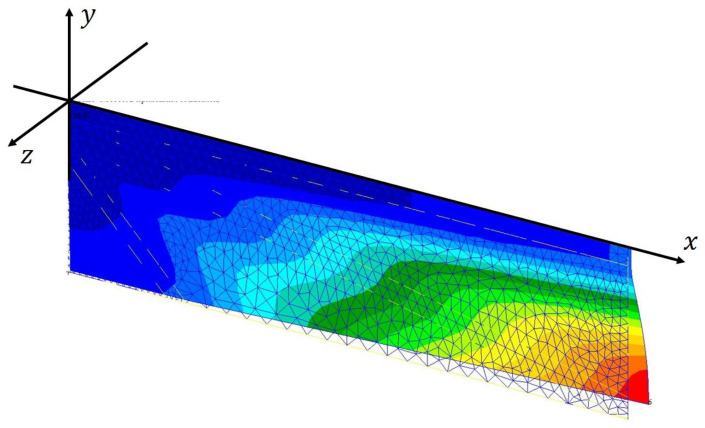
Deformation of the final design solution for our IMMW model, taking into account all fabrication conditions. The color profile shows the magnitude of the displacement in the z-direction.

**Figure 11 biomimetics-09-00133-f011:**
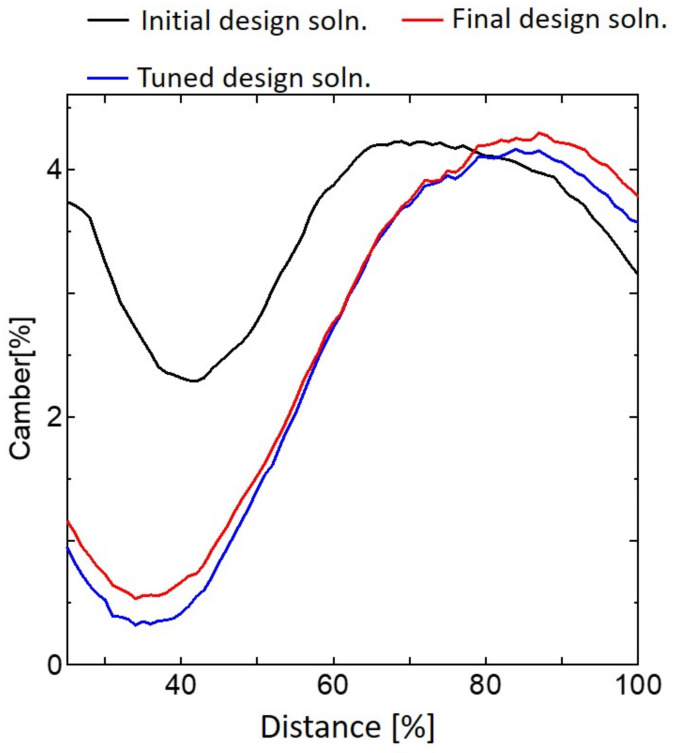
Camber distributions for the best design solution in Section 5.2, the tuned design solution, taking into account sheet adhesion, and the final design solution, taking into account all fabrication conditions.

**Table 1 biomimetics-09-00133-t001:** Comparison of CV average camber, maximum camber, and feathering angle for various CV thicknesses.

Thickness (μm)	Avg. Camber (%)	Max. Camber (%)	Feathering Angle (θ)
120	4.13	5.47	15.78
160	4.11	5.09	14.07
200	3.60	4.49	11.61

**Table 2 biomimetics-09-00133-t002:** Comparison of average camber mass of wing root vein for various root vein thicknesses.

Thickness (μm)	Avg. Camber (%)	Mass (mg)
160	3.50	1.96
200	3.69	1.97
240	4.06	1.98

## Data Availability

The datasets analyzed during the current study are available from the corresponding author upon reasonable request.

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
