# Peer review of "Novel Computational Design of Polymer Micromachined Insect-Mimetic Wings for Flapping-Wing Nano Air Vehicles"

_biomimetics, 2024, doi:10.3390/biomimetics9030133_

Round 1

Reviewer 1 Report (Previous Reviewer 2)

Comments and Suggestions for Authors

The paper is interesting as it has been globally developed and the information included are significant to justify the publication.

The paper is well-written in English.

References are adequate. Scientific soundness of present research is clear and the future applications are clearly presented.

For all these reasons I recommend to accept the publication.

Author Response

Dear Reviewer 1,

Many thanks for taking your time to review our manuscript. We wish to express our appreciation to you for giving us the recommendation for publishing our manuscript in its present form.

Sincerely,

Reviewer 2 Report (Previous Reviewer 3)

Comments and Suggestions for Authors

Thanks for the updated manuscript. I believe all my concerns from the previous submission has been addressed

Comments on the Quality of English Language

Minor editing of English language required

Author Response

Dear Reviewer 2,

Many thanks for taking your time to review our manuscript. We wish to express our appreciation to you for giving us the recommendation for publishing our manuscript with minor editing of English language.

About the point of minor editing English language, this manuscript has been checked and edited by a native English speaker at Forte, Inc., which is a Japan based company that employs editors and rewriters with a science background. Please kindly find the attached file entitled “R2304523-NativeCheckLetter.pdf”.

Thank you again for your valuable comments and suggestions that have led to significant improvement in the presentation and quality of this manuscript.

Sincerely,

Reviewer 3 Report (New Reviewer)

Comments and Suggestions for Authors

In this paper, the authors proposed a 2.5-dimensional insect-mimetic micro wing model for flapping-wing nano air vehicles. I recommend it can be accepted for publication after minor revision. My comments are as follows.

1.The number in Figure 4 is obscured by the content of the picture, the authors should adjust the position of the number.

2.In the part of Leading edge, the key data used by the authors to describe the thickness selection of Leading edge should be shown in Figure 6.

3.References are recommended to be updated. In addition, the format of many references is not standard and needs to be adjusted uniformly.

Comments on the Quality of English Language

Minor editing of English language required for all text.

Author Response

Dear Reviewer 3,

Many thanks for taking your time to review our manuscript. We wish to express our appreciation to you for giving us the recommendation for publishing our manuscript after minor revision.

About our point-by-point replies to your valuable comments and suggestions, please see the attached file entitled “Authors_reply_letter-2858877_Feb17_2024 plus R2304523-NativeCheckLetter.pdf”

About the point of minor editing English language for all text, this manuscript has been checked and edited by a native English speaker at Forte, Inc., which is a Japan based company that employs editors and rewriters with a science background. Please kindly find the letter from this company which is attached to the last page of the above file.

Thank you again for your valuable comments and suggestions that have led to significant improvement in the presentation and quality of this manuscript.

Sincerely,

This manuscript is a resubmission of an earlier submission. The following is a list of the peer review reports and author responses from that submission.

Round 1

Reviewer 1 Report

Comments and Suggestions for Authors

This paper “Novel computational design of polymer micromachined insect mimetic wings for flapping-wing nano air vehicles” describes a 2.5-dimensional insect-mimetic micro wing model for flapping-wing nano air vehicles The proposed shell element model consists of a wing membrane, a leading edge, a center vein, and a root vein that can be fabricated using polymer micromachining. A parametric study (a combination of wing membrane, wing leading edge, wing center edge, and wing root edge) has been conducted to search for a good wing design closest to a real insect wing. However, there are some flaws and unclear points that need to be addressed before accepting for publication.

1.      You have mentioned the wing dimension is from an actual insect wing, what kind of insect wing did you refer to for your wing design?

2.      The wing is not uniform, how do you calculate the equivalent E and I in equation (3)?

3.      The actual insect wings experience not only aerodynamic forces (lift, drag, moment) but also wing inertia (which plays an important role in wing deformation) during flapping. However, you only consider drag force in your wing design, why do you drop out the other force? I do not think the drag is only the most dominant force in insect flight.

4.      In your finite element analysis, you need to provide more details about shell element type, model meshing, time step, etc., for readers to understand.

5.      What location did you calculate your wing camber? And can you provide wing camber at different wing sections (i.e., 0%, 25%, 50% 75%, and 100% wing length) and compared to the actual insect wing?

6.      Your final wing design has a feathering angle of about 11.13deg. I do not see it is comparable to the one of real insect wing of about 45deg.

7.      The real insect wing experiences not only bending force but also torsion, did you consider the torsional stiffness of your proposed wing design?

8.      You mentioned your wing design which considers the feasible fabrication process, but I do not see how your fabricated wing is. Have you tried to fabricate your proposed wing and flap it at a given frequency comparable to the insect wing you mimic?

Reviewer 2 Report

Comments and Suggestions for Authors

The paper is interesting as it has been globally developed and the information included are significant to justify the publication.

The paper is well-written in English.

References are adequate. Scientific soundness of present research is clear and the future applications are clearly presented.

For all these reasons I recommend to accept the publication.

Reviewer 3 Report

Comments and Suggestions for Authors

This work proposed a biomimetic micro wing model and then conducted investigation of the modeling and iteration of design optimization. Overall, the quality of this work is good, and it should meet the acceptance requirement after the comment below regarding the novelty of this work is properly solved.

·      Literature review should be conducted with more recent works in the field, currently the cited references are overall outdated. Also, there seems to be several preliminary works done by the authors based on the cited publications, and this work is solely a model simplification, which means the novelty contribution may be relatively low.

Comments on the Quality of English Language

minor revision needed
